# Genome-scale metabolic reconstruction of the symbiosis between a leguminous plant and a nitrogen-fixing bacterium

George C. diCenzo [1,2], Michelangelo Tesi[1], Thomas Pfau[3], Alessio Mengoni [1✉] & Marco Fondi [1✉]

The mutualistic association between leguminous plants and endosymbiotic rhizobial bacteria is a paradigmatic example of a symbiosis driven by metabolic exchanges. Here, we report the reconstruction and modelling of a genome-scale metabolic network of *Medicago truncatula* (plant) nodulated by *Sinorhizobium meliloti* (bacterium). The reconstructed nodule tissue contains five spatially distinct developmental zones and encompasses the metabolism of both the plant and the bacterium. Flux balance analysis (FBA) suggests that the metabolic costs associated with symbiotic nitrogen fixation are primarily related to supporting nitrogenase activity, and increasing $N_2$-fixation efficiency is associated with diminishing returns in terms of plant growth. Our analyses support that differentiating bacteroids have access to sugars as major carbon sources, ammonium is the main nitrogen export product of $N_2$-fixing bacteria, and $N_2$ fixation depends on proton transfer from the plant cytoplasm to the bacteria through acidification of the peribacteroid space. We expect that our model, called 'Virtual Nodule Environment' (ViNE), will contribute to a better understanding of the functioning of legume nodules, and may guide experimental studies and engineering of symbiotic nitrogen fixation.

---

[1] Department of Biology, University of Florence, Sesto Fiorentino, Italy. [2] Department of Biology, Queen's University, Kingston, ON, Canada. [3] Life Sciences Research Unit, University of Luxembourg, Belvaux, Luxembourg. ✉email: alessio.mengoni@unifi.it; marco.fondi@unifi.it

Macroorganisms are colonized by a staggering diversity of microorganisms, collectively referred to as a 'holobiont'[1,2]. The intimate association between organisms is often driven by metabolic exchanges: many insects obtain essential nutrients from obligate bacterial symbionts[3], most plants can obtain phosphorus from arbuscular mycorrhiza in exchange for carbon[4], and the gut microbiota is thought to contribute to animal nutrition[5,6]. Complex global patterns often emerge during these intimate biological associations[7], especially when nutritional inter-dependencies are involved[8–10]. The communication between the two metabolic networks of the interacting organisms may give rise to unpredicted phenotypic traits and unexpected emergent properties. Metabolic relationships can span over a large taxonomic range and have profound biological relevance[11–14]. For example, the interactions between bacteria and multicellular organisms have been suggested to be key drivers of evolutionary transitions, leading to eukaryotic diversification and to the occupancy of novel niches[9,15,16]. The study of the association of two biological entities is mainly challenged by the size of the system and by the unpredictability of their metabolic interactions. Theoretical, systems-level models are required to unravel the intimate functioning of metabolic associations and to eventually exploit their potential in biotechnological applications.

Symbiotic nitrogen fixation (SNF) is a paradigmatic example of the importance and the complexity of natural biological associations. SNF is a mutualistic relationship between a group of plant families, including the Fabaceae, and a polyphyletic group of alpha- and beta-proteobacteria known as rhizobia, or a taxa of Actinobacteria (*Frankia* spp.), in which the plants provide a niche and carbon to the bacteria in exchange for fixed nitrogen[17]. SNF involves constant metabolic cross-talk between the plant and the bacteria[18], and it is a paradigmatic example of bacterial cellular differentiation[19] and sociomicrobiological interactions[20]. The rhizobia intra-cellularly colonize plant cells of a specialized organ known as a root (or stem) nodule. The intra-cellular rhizobia (referred to as bacteroids) are surrounded by a plant-derived membrane, and the term symbiosome is used in reference to the structure consisting of the bacteroid, the plant-derived membrane (i.e., the peribacteroid membrane), and the intervening space (i.e., the peribacteroid space). Nodules with an indeterminate structure, such as those formed by the plant *Medicago truncatula*, are divided into spatially distinct developmental zones[21] with a distal apical meristem and a proximal nitrogen fixation zone.

SNF plays a key role in the global nitrogen cycle and is central to sustainable agricultural practices by reducing the usage of synthetic nitrogen fertilizers whose application results in a multitude of adverse environmental consequences[22–24]. Unfortunately, our ability to maximize the benefit of SNF is limited since rhizobial inoculants are often poorly effective due to low competitiveness[25,26] and because rhizobium symbioses are specific to leguminous plants. Manipulating the rhizobium – legume interaction for biotechnological purposes will require an in-depth understanding of the symbiotic interaction, as well as an ability to predict the consequences of genetic changes and environmental perturbations.

From a metabolic perspective, genome-scale metabolic reconstruction (GENREs) and constraint-based modeling has great potential to fulfill these roles. A GENRE also serves as a comprehensive knowledgebase of an organism's metabolism, containing hundreds to thousands of metabolic and transport reactions, most of which are linked to the corresponding gene(s) whose gene product(s) catalyzes the reaction[27,28]. With the aid of mathematical approaches such as flux balance analysis (FBA), GENREs can be used to identify emergent system-level properties, to predict active reactions, and to identify essential genes[29]. Compared to simple enrichment analyses that are typical in omics studies, GENRE-based methods allow for the interpretation of data in a connected manner based on network topology and to infer the effects of changes in remote pathways on the overall cell physiology. When considering interacting entities, for example, this approach can predict the consequence of mutations in one organism on the metabolism of the other. However, multi-organism metabolic reconstructions are still in their infancy, and very few examples of combined models exist compared to single strain GENREs[8,14,30–32].

Despite the importance of metabolism to SNF[18], there has been limited use of metabolic modeling in the study of rhizobia and SNF. To date, GENREs of varying quality have been reported for only three rhizobia: *Sinorhizobium meliloti*[33–35], *Rhizobium etli*[36–38], and *Bradyrhizobium diazoefficiens*[39]. Currently, *M. truncatula*[40] and *Glycine max*[41] are the only legumes with published GENREs. With the exception of the *G. max* GENRE, these GENREs have been used in preliminary analyses of SNF, providing results generally consistent with expectations. However, all analyses to date suffer from two major limitations. Simulations with the rhizobium models ignore plant metabolism, while simulations with the *M. truncatula* GENRE (based on the genome sequence published in 2011[42], which has since been updated in 2014[43] and again 2018[44]) involved a very limited draft *S. meliloti* metabolic reconstruction (whose genome sequence was published in 2001[45]). Furthermore, all simulations have focused on the final stage of SNF and have not considered the different steps of the preceding developmental progression where metabolism remains poorly understood[18].

Here, we report a holistic in silico representation of the integrated metabolism of the holobiont consisting of a *M. truncatula* plant nodulated by *S. meliloti*, which we refer to as a <u>Vi</u>rtual <u>N</u>odule <u>E</u>nvironment (ViNE). Our combined, multi-compartment reconstruction accounts for the metabolic activity of shoot and root tissues together with a nodule consisting of five developmental zones. We report initial characterizations of ViNE using FBA, including zone-specific metabolic properties, trade-offs between nitrogen-fixation and plant growth, and the usage of dicarboxylates as a carbon source by bacteroids. Going forward, we expect ViNE will provide a powerful platform for hypothesis generation aimed at understanding and quantitatively evaluating SNF, as well as guiding attempts at engineering SNF for increased symbiotic efficiency.

## Results

**Validation of iGD1348, an updated *S. meliloti* reconstruction.** Prior to constructing the integrated plant – bacterium metabolic model, an updated metabolic reconstruction of *S. meliloti* Rm1021 was prepared as described in Methods and Supplementary Note 1. Briefly, the highly refined core metabolic network iGD726[34] was combined with the comprehensive accessory metabolism of the genome-scale metabolic network iGD1575[33]. Most of the reactions were manually curated through comparison against the literature, referenced where possible, and mass and charge balanced. The updated model consists of 1348 genes (Supplementary Table 1), and incorporates information from 240 literature sources (Supplementary Data 1) that includes transposon-sequencing (Tn-seq) data[34] and Phenotype Micro-Array data[33,46,47] for wild-type and mutant strains. The reduced size of iGD1348 relative to the older iGD1575 model is a consequence of deleting poorly characterized genes that appeared to be incorrectly associated with reactions, and the removal of pathways that produced dead-end metabolites.

Several tests were performed to validate the quality of the newly prepared *S. meliloti* reconstruction. Flux balance analysis (FBA) was used to simulate growth using glucose or succinate as the sole source of carbon, with or without the inclusion of an NGAM reaction. Inclusion of an NGAM reaction resulted in a specific growth rate reduction of ~0.043 h$^{-1}$ and 0.030 h$^{-1}$ for

growth with glucose and succinate, respectively. This result confirmed the absence of energy leaks in iGD1348 that would allow for spontaneous energy production.

Using FBA, the ability of *S. meliloti* to catabolize 163 carbon sources to support growth was predicted with the iGD1348 and iGD1575 models (Supplementary Data 2). As previously reported[33], simulations with the iGD1575 model correctly predicted growth with 67 of the 85 (79%) substrates experimentally shown to support growth of *S. meliloti*. Nicely, simulations with the iGD1348 model correctly predicted growth with 76 of these 85 (89%) substrates, including all 67 that supported growth of the iGD1575 model. This result confirmed that iGD1348 incorporates the majority of the accessory metabolism of *S. meliloti*, and that it is a better representation of total cellular *S. meliloti* metabolism than the previous genome-scale model.

Context-specific core metabolic models, containing a minimal set of reactions for biomass production in a given environment, were extracted from the iGD1348 and iGD1575 metabolic models through the integration of Tn-seq data[34] using Tn-Core[48]. The accuracy of the resulting core metabolic models was determined through comparison with iGD726, a manually prepared core metabolic model of *S. meliloti*[34]. As summarized in Fig. 1, the iGD1348 core model displayed greater overlap with the iGD726 model than did the core model generated from iGD1575 (Fig. 1a). In particular, of the genes essential in the iGD726 model, 96% were essential in the iGD1348 core model, whereas only 62% were essential in the iGD1575 core model (Fig. 1b). This result confirmed that the newly prepared iGD1348 reconstruction better represents the core metabolic network of *S. meliloti* than does the iGD1575 reconstruction. Overall, these tests confirmed that iGD1348 is a high-quality representation of *S. meliloti* metabolism, and that it is an improvement over previous iterations of *S. meliloti* GENREs.

## Construction and validation of a model of a nodulated legume.
Obtaining high-quality reconstructions of *M. truncatula* and *S. meliloti* metabolism was a prerequisite to generating an in silico genome-scale metabolic network of an entire nodulated legume (referred to as ViNE). For *S. meliloti*, we used the updated model described in the previous section. In the case of *M. truncatula*, we used a recently published reconstruction that was updated to match the most recent version of the *M. truncatula* genome annotation (see Supplementary Note 2).

Integrating the *S. meliloti* and *M. truncatula* metabolic models resulted in a model encompassing shoot, root, and nodule tissues as summarized in Fig. 2 and Table 1. Memote[49] was used to evaluate the overall quality of the integrated reconstruction. This analysis revealed an overall quality score of 72%, which falls well in the range of the scores of the original two models (78% and 65% for the iGD1348 and the *M. truncatula* reconstructions, respectively) and is consistent with ViNE being a high-quality metabolic reconstruction. The detailed reports are provided in Supplementary Data 3.

In total, ViNE includes 746 unique *S. meliloti* genes and 1,327 unique *M. truncatula* genes. The nodule was subdivided into five zones to match the spatially, and transcriptionally, distinct developmental zones that are simultaneously present in indeterminate legume nodules such as those formed by *M. truncatula*[21,50]. Several simulations were performed to evaluate the reliability of the model. Using FBA, the maximal rate of plant (shoot + root) growth of the nodulated system was predicted to be ~0.044 g day$^{-1}$ (g plant dry weight)$^{-1}$, while the N$_2$-fixation rate was predicted to be ~3 µmol h$^{-1}$ (g plant dry weight)$^{-1}$. Both values are reasonable predictions; *Medicago sativa* plants have an experimentally determined growth rate of ~0.1 g day$^{-1}$ (g plant

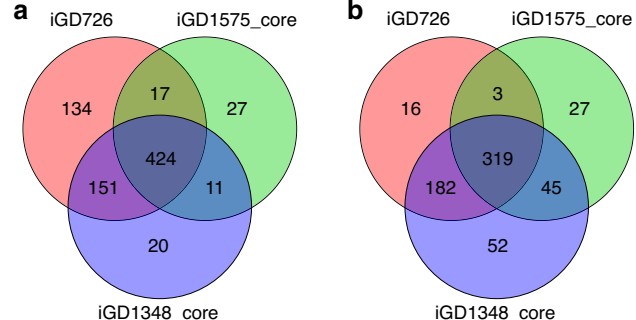

**Fig. 1 Overlap between the iGD726 model and core metabolic models derived from iGD1575 and iGD1348.** Venn diagrams illustrating the overlap in (**a**) the total gene content, and (**b**) the essential genes of the following three models: the manually prepared iGD726 core model, a core model derived from iGD1575, and a core model derived from iGD1348. Core models of iGD1575 and iGD1348 were prepared using Tn-Core and published Tn-seq data.

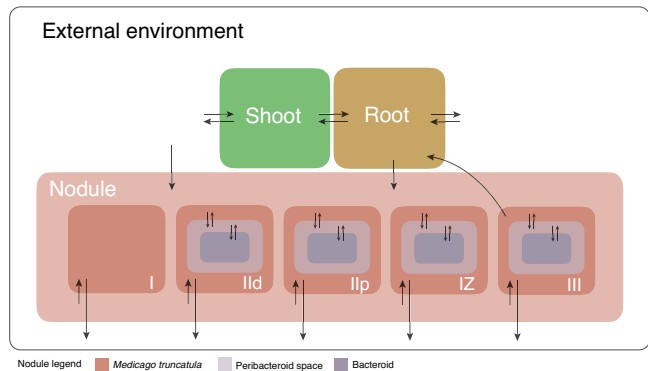

**Fig. 2 Visual depiction of ViNE.** A schematic summarizing the overall structure of the *S. meliloti* nodulated *M. truncatula* plant developed in this work. The model contains three plant tissues (shoot, root, nodule) with the nodule subdivided into five developmental zones (I, IId, IIp, IZ and III). Arrows indicate transport reactions with the direction representative of the directionality of the transport reactions. The scale of the figure has no meaning.

dry weight)$^{-1}$ [51] and N$_2$-fixation rates around 7 µmol h$^{-1}$ (g plant dry weight)$^{-1}$ based on acetylene reduction assays[52]. Moreover, the predicted carbon cost of supporting N$_2$-fixing nodules (nodule CO$_2$ export per N$_2$ fixed) was ~4.2 g C g$^{-1}$ -N, falling within the normal range of 3–5 g C g$^{-1}$ -N based on experimental measurements[53].

Next, FBA was used to examine the effects of adding exogenous ammonium to the soil on plant growth considering two situations: (i) the N$_2$-fixation efficiency (defined as the rate of N$_2$-fixation per gram nodule) could vary while the rate of nodulation (defined as the percent of total plant biomass being nodule biomass) was constant at 2%, and (ii) the rate of nodulation could vary while the N$_2$-fixation efficiency was constant at 150 µmol h$^{-1}$ (g nodule dry weight)$^{-1}$. As expected, increasing the availability of exogenous ammonium increased the rate of plant growth, with the effect more pronounced when the rate of nodulation was allowed to decrease with increasing ammonium since the plant no longer had to invest in nodule maintenance (Fig. 3).

We then simulated the effects of individual bacteria gene deletions on plant biomass production (Supplementary Data 4)

**Table 1 Summary properties of ViNE.**

| Model feature | Shoot | Root | Zone I | Zone IId | Zone IIp | Zone IZ | Zone III |
|---|---|---|---|---|---|---|---|
| Genes | | | | | | | |
| *M. truncatula* | 1295 | 1292 | 236 | 265 | 243 | 228 | 116 |
| *S. meliloti* | 0 | 0 | 0 | 640 | 629 | 638 | 246 |
| Reactions[a] | | | | | | | |
| *M. truncatula* | 937 | 944 | 494 | 543 | 559 | 530 | 318 |
| *S. meliloti* | 0 | 0 | 0 | 662 | 654 | 670 | 201 |
| Metabolites | | | | | | | |
| *M. truncatula* | 831 | 825 | 490 | 568 | 597 | 581 | 286 |
| *S. meliloti* | 0 | 0 | 0 | 751 | 747 | 764 | 222 |

[a]Excludes reactions for transfer of metabolites between tissues or between *M. truncatula* and the peribacteroid space.

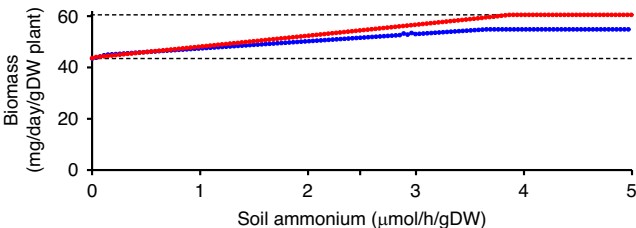

**Fig. 3 Effect of exogenous ammonium on *M. truncatula* growth.** The effects of increasing the availability of soil ammonium on the growth rate of nodulated *M. truncatula* was examined. Simulations were run allowing either the $N_2$-fixation efficiency to vary while the rate of nodulation remained constant at 2% (blue) or allowing the rate of nodulation to vary while the $N_2$-fixation efficiency remained constant at 150 μmol h$^{-1}$ (g nodule dry weight)$^{-1}$ (red). The dashed lines indicate the maximal rate of plant growth with exogenous ammonium (upper) and the maximal rate of plant growth when relying on $N_2$-fixation (lower).

and compared the results to published experimental data. The model was able to accurately predict the phenotypes of many *S. meliloti* mutants. For example, *S. meliloti* genes such as *nifH* (nitrogenase), *dctA* (succinate transport), *ilvI* (branched-chain amino acid biosynthesis), *aatA* (aspartate transaminase), *pgk* (phosphoglycerate kinase), and *nrdJ* (ribonucleotide reductase) were correctly predicted to be essential, while *pyc* (pyruvate carboxylase), *glnA* (glutamine synthetase), *pckA* (phosphoenol-pyruvate carboxykinase), and *leuB* (leucine biosynthesis) were correctly predicted to be non-essential[54–61]. Similarly, the removal of plant-encoded nodule sucrose synthase, phosphoe-nolpyruvate carboxylase, and homocitrate synthase reactions abolished nitrogen fixation, as expected[62–64]. However, it is important to note that the predictions were not perfect. For example, deleting *argG* (arginine biosynthesis) or *carA* (carbamoyl phosphate synthase) did not result in the expected phenotypes, while the incorrect malic enzyme (*tme* instead of *dme*) was predicted to be essential[65–67]. Taken together, these analyses provide support for the general reliability of ViNE as a representation of nodule metabolism.

**Metabolic progression and nutrient exchange during nodule development.** The presence of five nodule zones in ViNE provided an opportunity to examine the metabolic changes associated with the development of an effective nodule. To accomplish this, FBA was used to predict the flux distribution through the integrated metabolic networks of each nodule zone, and to simulate the effects of individually deleting each gene, or removing each reaction, specifically in a single nodule zone.

Additionally, a robustness analysis was performed to evaluate how perturbations in the flux of individual bacteroid reactions influence the predicted rate of plant growth. The outputs of these analyses are provided as Supplementary Data 5 and 6, and they are summarized in Fig. 4 and Supplementary Figs. S1 and S2. For simplicity, here we focus on the reaction-level analyses, and we split the nodule into only three sections: uninfected (zone I), differentiating (zones IId, IIp, and IZ), and nitrogen-fixing (zone III). Highlighting the overall similarity of zones IId, IIp, and IZ, and thus supporting their grouping, the robustness analysis indicated that roughly 90% of the bacteroid reactions that had to carry flux in one of these zones had to carry flux in all three zones to maximize plant growth.

The most notable difference comparing the uninfected and differentiation zones was an increase in the number of active reactions related to energy production, including carbon and nucleotide metabolism. This result suggests that the accommodation of differentiating bacteroids may place additional energy demands on the plant cell, and that few additional metabolic functions are required. In contrast, the transition from the differentiation zone to the $N_2$-fixing zone was associated with a marked decrease in the number of active reactions in both the plant and bacterial cells, consistent with published transcriptomic and proteomic datasets[50,68–70]. Highlighting this result, ~ 560 bacteroid reactions had to carry flux in the differentiation zones to optimize plant growth, whereas only 167 bacteroid reactions had to carry flux in the $N_2$-fixation zone for maximal plant growth.

The lack of biomass production in the $N_2$-fixation zone meant that most biomass biosynthetic pathways were predicted to be inactive and non-essential. However, bacterial pathways related to the production of cofactors for nitrogenase or energy production remained essential; this included FMN, heme, cobalamin, pyridoxine phosphate, and glutathione biosynthesis, as well as the pentose phosphate pathway. Similarly, the TCA cycle, oxidative phosphorylation, and purine biosynthesis in bacteroids were predicted to be essential in the $N_2$-fixation zone, presumably to supply the massive amounts of energy required by nitrogenase. Biosynthesis of methionine and SAM were also predicted to be essential. Few other notable bacterial reactions were required in the $N_2$-fixation zone (Supplementary Data 5). In the plant compartment, the majority of the active reactions were related to central carbon metabolism for the production of energy and $C_4$-dicarboxylates for use by the bacteroids, while other active reactions were involved in the assimilation of ammonium through the formation of glutamine. Consistent with experimental works [reviewed by[18,71]], the FBA results indicated that the plant nodule cells are provided sucrose as a carbon/energy source; in fact, ~ 30% of all carbon fixed by the plant leaves was sent to nodule zone III. The sucrose was then hydrolyzed and metabolized to phosphoenolpyruvate, of which ~ 80% was

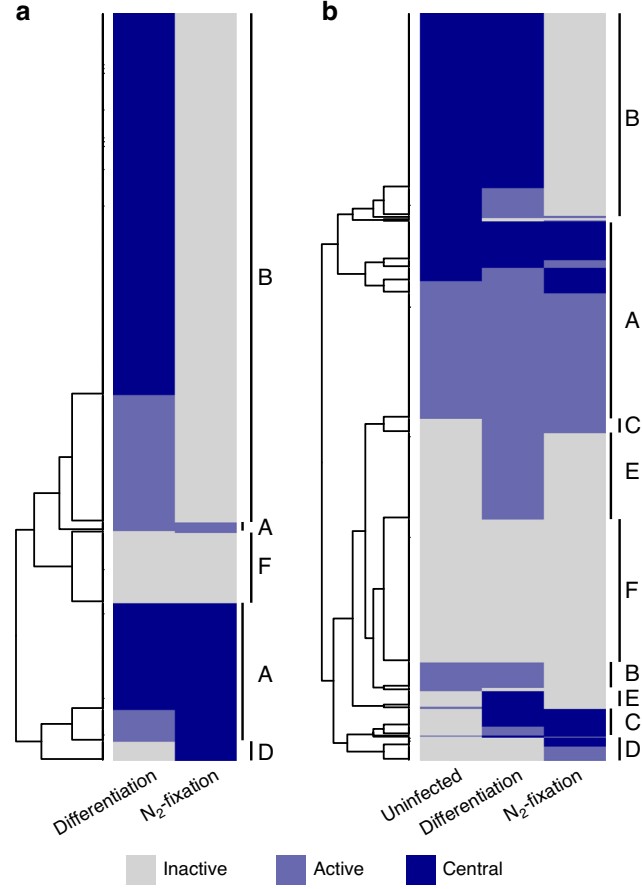

**Table 2 Contributions of N$_2$-fixation and nodulation to the metabolic costs of SNF.**

| Nitrogen source | Nodulation state | Relative plant growth rate |
|---|---|---|
| N$_2$-fixation | Nodulated | 0.717 |
| N$_2$-fixation | Nodulated but without maintenance costs | 0.781 |
| N$_2$-fixation | Non-nodulated[a] | 0.812 |
| Exogenous ammonium | Non-nodulated[a] | 1 |

[a]Non-nodulated indicates that nodule biomass was excluded from the overall biomass reaction and nodule maintenance costs were removed. As N$_2$-fixation in ViNE is unlinked to the production or maintenance of nodule biomass (unless stated otherwise), N$_2$-fixation can occur without nodulation.

**Fig. 4 Nodule zone-specific metabolism.** Heatmaps are presented displaying which reactions are inactive (gray), active but whose removal does not impair plant growth (light blue), or central (dark blue; growth reduction > 10% for the model missing the reaction compared to the full model) in the different nodule zones. In the differentiation zone, reactions are marked as central only if it was central in each of zone IId, IIp, and IZ. Heatmaps are shown for (**a**) *S. meliloti* reactions and (**b**) *M. truncatula* reactions. Reactions were clustered using hierarchical clustering, and the following main clusters were identified: A – constitutively active; B – specific to growing cells; C – specific to infected cells; D – specific to the nitrogen-fixation zone; E – specific to the differentiation zone; F – constitutively inactive.

diverted to oxaloacetate through a cytoplasmic phosphoenolpyruvate carboxykinase reaction for use in the production of C$_4$-dicarboxylates.

Next, nutrient exchange between the plant and bacterial partners was examined. While the prevailing evidence suggests C$_4$-dicarboxylates (succinate, malate, fumarate) are the primary carbon source for N$_2$-fixing bacteroids[18,72–74], the source of carbon for differentiating bacteroids has not been established. The FBA results suggested that differentiating bacteroids primarily use sugars, likely sucrose, as a carbon source. This is consistent with micrographic evidence suggesting that bacterial mutants unable to use C$_4$-dicarboxylates can undergo at least partial differentiation[72,73]. Currently, it is commonly accepted that nitrogen is primarily exported from bacteroids as ammonia[75,76]; however, some studies have suggested that L-alanine could be a major nitrogen export product[77,78]. Our FBA simulations were consistent with ammonia being the primary export product in the *S. meliloti* – *M. truncatula* symbiosis. However, prior to constraining the nodule reaction space, reducing the availability of oxygen to the bacteroids resulted in

a shift in the nitrogen export product from ammonia to L-alanine. Thus, the detection of L-alanine versus ammonia as an export product could be due, in part, to differences in experimental set-up that may influence oxygen availability to the bacteroid. Also, experimental data suggest that rhizobial biosynthesis of some amino acids is essential for the symbiosis while the biosynthesis of others is not, and that the phenotypes may be symbiosis-specific [reviewed by[79]]. Similarly, our FBA simulations suggested that rhizobial biosynthesis of approximately half of the amino acids was essential for the symbiosis.

**Sociobiology of symbiotic nitrogen fixation.** By containing a representation of an entire nodule, ViNE allowed for an evaluation of the metabolic costs associated with SNF. Using FBA, the maximal plant growth rate of the nodulated system (without exogenous ammonium) was estimated to be ~72% of the maximal growth rate of a nodule-free system supplied with non-limiting amounts of exogenous ammonium (Fig. 3, Table 2). The largest factor contributing to the difference in growth was the direct energetic cost of supporting N$_2$-fixation (~67% of the difference; Table 2). The remaining third of the difference was explained by the cost of synthesizing (~11% of the difference) and maintaining (~22% of the difference) the nodule and bacteroid tissue (Table 2).

We next evaluated the relationship between N$_2$-fixation efficiency (while keeping the rate of nodulation at 2%) and the rate of plant growth. When the N$_2$-fixation efficiency was below the optimum, there was a linear relationship between N$_2$-fixation and biomass production (Fig. 5a). However, excessive N$_2$-fixation quickly resulted in impaired plant growth, with a 10% excess of N$_2$-fixation collapsing the symbiosis (Fig. 5a). We hypothesized that this result was due to insufficient energy to support both the excess N$_2$-fixation and the ATP maintenance costs. Consistent with this hypothesis, removing the upper limit on the rate of zone III oxygen uptake resulted in a gradual decrease in plant growth as the rate of N$_2$-fixation was increased above the optimal (Fig. 5a). In this case, excessive N$_2$-fixation was less detrimental than insufficient N$_2$-fixation; the effect of increasing the N$_2$-fixation efficiency by 50 µmol h$^{-1}$ (g nodule dry weight)$^{-1}$ increased or decreased the rate of plant growth by 14.7 or 3.4 mg day$^{-1}$ (g plant dry weight)$^{-1}$ when below or above the optimum, respectively. We next examined the consequences of varying the rate of nodulation (i.e., the ratio between plant and nodule biomass) while maintaining a constant N$_2$-fixation efficiency. The simulations demonstrated linear relationships between the rate of nodulation and plant growth both above and below the optimum (Fig. 5b), with increasing the percent nodulation resulting in a 3-fold greater impact when below the optimum compared to above the optimum. Overall, these

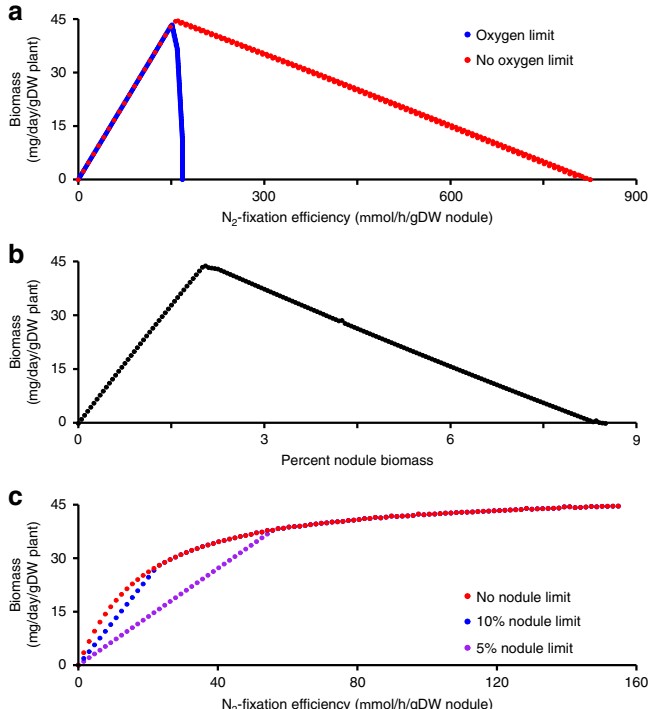

**Fig. 5 Relationships between plant growth and rate of N$_2$-fixation or nodulation.** In all panels, the sole source of nitrogen for the plant is through N$_2$-fixation. **a** Pareto frontiers showing the relationship between the N$_2$-fixation efficiency (with a constant rate of nodulation of 2%) and the rate of plant biomass production using ViNE with default parameters (blue) or no limit on zone III oxygen usage (red). **b** The relationship between the rate of nodulation and the rate of plant biomass production (with a constant N$_2$-fixation efficiency of 150 μmol h$^{-1}$ (g nodule dry weight)$^{-1}$). **c** The effect of the N$_2$-fixation efficiency on the rate of plant growth, with the amount of nodule biomass optimized to maximize plant growth and without a limit on zone III oxygen uptake (see Supplementary Fig. 3 for simulations with an oxygen uptake limit). Nodule biomass was either uncapped (red) or limited to 10% (blue) or 5% (purple) of the overall plant + nodule biomass.

simulations suggest that a slightly too efficient symbiosis is preferable (for plant biomass production) over a slightly inefficient symbiosis, unless the required rate of O$_2$ usage exceeds the nodule oxygen diffusion limit.

The previous simulations represented simple scenarios where only a single variable differed. In reality, a change in the N$_2$-fixation efficiency should be accompanied by a change in the rate of nodulation as a result of legume autoregulation of nodulation[80]. We therefore ran simulations where the N$_2$-fixation efficiency was varied and the rate of nodulation was optimized to maximize plant growth. Strikingly, the simulations suggested a pattern of diminishing returns associated with increasing the N$_2$-fixation efficiency (Fig. 5c and Supplementary Fig. 3); decreasing the N$_2$-fixation efficiency 50% from the maximum tested value resulted in a mere 10% decrease in plant growth. The half-maximal growth rate was achieved with a N$_2$-fixation efficiency of just 10% the maximal, although this required that the nodule accounted for almost 13% of the total biomass. If we assume an upper limit of nodulation at 10% or 5% of the total biomass, the benefits of low N$_2$-fixation efficiencies are decreased although the pattern of diminishing returns remains (Fig. 5c and Supplementary Fig. 3). In these cases, half-maximal plant growth rate is achieved at 12% or 21%, respectively, of the highest tested N$_2$-fixation efficiency. Overall, these simulations support that even a poor symbiosis is likely to provide a noticeable benefit to the plant.

**H$^+$ and O$_2$ influence on the carbon source provided to bacteroids.** It is well-established that C$_4$-dicarboxylates (malate, succinate, fumarate) are the primary carbon source provided to nitrogen-fixing zone III bacteroids[18,72,73]; however, the reason for this remains unclear. We therefore attempted to uncover a metabolic explanation using ViNE. Surprisingly, preliminary FBA simulations with the ViNE precursor model (i.e., prior to constraining the reaction space) suggested that the N$_2$-fixing bacteroids of nodule zone III are provided sucrose, not C$_4$-dicarboxylates, as the primary carbon source. Unexpectedly, forcing the use of C$_4$-dicarboxylates resulted in the model being unable to fix nitrogen or produce plant biomass. During those simulations, protons of the plant cytosol could be transferred to the peribacteroid space but were not allowed to be used by the N$_2$-fixing bacteroids. However, this may not be realistic since the peribacteroid space of N$_2$-fixing bacteroids is acidic due to import of protons from the plant cytosol[81–83]. If the analysis was repeated and the zone III bacteroids were provided access to the protons of the peribacteroid space, it became possible for C$_4$-dicarboxylates to serve as the primary carbon source and support N$_2$-fixation and plant growth. These results suggest that the plant-driven acidification of the peribacteroid space is essential for the metabolic functioning of the bacteroid.

Although the transfer of protons to the periplasm allowed C$_4$-dicarboxylates to support N$_2$-fixation, the rate of plant biomass production nevertheless remained higher when the N$_2$-fixing bacteroids were provided sucrose instead of C$_4$-dicarboxylates. To further investigate this difference, ViNE was modified to contain reactions for the transport and metabolism of sucrose by N$_2$-fixing bacteria (see Supplementary Note 3). Consistent with results from the precursor model, FBA simulations suggested the ability of bacteroids to use sucrose (plus C$_4$-dicarboxylates) increased the plant growth rate by 6.4% relative to when bacteroids were supplied only C$_4$-dicarboxylates. ViNE contains a limit on the rate of oxygen uptake by zone III nodule tissue as the concentration of free oxygen in the N$_2$-fixation zone is known to be low[84] to protect the highly oxygen-sensitive nitrogenase enzyme. The expected consequence of this is a constriction in the rate of flux through the electron transport chain, thus, restricting nodule and bacteroid metabolism by limiting ATP production and the removal of reductant. We wondered whether the use of sucrose versus C$_4$-dicarboxylates may be modulated by the free oxygen concentration of the nodule. The concentration of free oxygen in the N$_2$-fixation zone has been experimentally demonstrated to be <50 nM[84]. Notably, the K$_m$ values of the mitochondrial and bacterial terminal oxidases towards oxygen are 50−100 nM[85,86] and 7 nM[87], respectively. These enzyme kinetics suggest that the metabolism of the plant fraction, but not the bacteroid fraction, of the nodule is likely to be oxygen limited[88,89], a conclusion that is supported by measurements of nodule adenylate pools[90]. Therefore, we ran a series of simulations in which the upper limit of the mitochondrial terminal oxidase reaction of zone III was varied, with no overall limit on the use of oxygen by the nodule. Gradually reducing the flux through the mitochondrial terminal oxidase was associated with a gradual replacement of sucrose with C$_4$-dicarboxylates as the carbon source provided to N$_2$-fixing bacteroids (Fig. 6). This result is consistent with the hypothesis that the low free oxygen concentration of the N$_2$-fixation zone could be a contributing factor as to why bacteroids are provided C$_4$-dicarboxylates, and not sugars, as the primary carbon source.

Assuming the nodule (consisting primarily of zone III tissue) accounts for 2% of total plant biomass, and that bacteroid biomass accounts for 25% of nodule biomass, the maximal rate of predicted C$_4$-dicarboxylate import by N$_2$-fixing bacteroids (1.3 mmol h$^{-1}$ [g bacteroid dry weight]$^{-1}$) was similar to

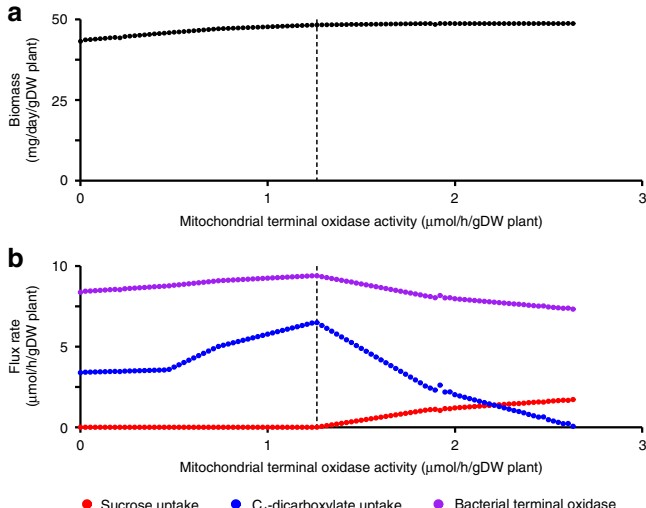

**Fig. 6 Effects of limiting the mitochondrial terminal oxidase of the N₂-fixing zone.** FBA simulations were run with a modified version of ViNE in which bacteroid metabolism could be supported by sucrose in addition to $C_4$-dicarboxylates. No overall limit on oxygen usage of the nodule was set during the simulations, but a limit was set on the activity of the mitochondrial terminal oxidase of the zone III nodule tissue. **a** The effect on plant growth rate of varying the mitochondrial terminal oxidase of the zone III nodule tissue. **b** The effect on specified flux rates of varying the mitochondrial terminal oxidase of the zone III nodule tissue. Red – the sucrose uptake rate of N₂-fixing bacteroids; blue – the uptake rate of $C_4$-dicarboxylates of N₂-fixing bacteroids; purple – the flux rate of the terminal oxidase of N₂-fixing bacteroids. The dashed line indicates the mitochondrial terminal oxidase flux rate below which no sucrose is used by the bacteroids.

experimentally determined uptake rates by *S. meliloti* bacteroids (1.1 to 1.3 mmol h⁻¹ [g bacteroid dry weight]⁻¹)[35,91]. Considered together, these simulations provide evidence that $C_4$-dicarboxylates can support optimal plant growth under physiologically relevant conditions.

## Discussion

Models of the integrated metabolism of various holobionts (consisting of a host and its symbiotic microorganisms) would be valuable tools to understand the emergent properties of these systems[92,93]. However, to date there are few examples of constraint-based metabolic modeling being used to study metabolic interactions [e.g.[8,31]], with this approach most commonly used to study the human gut microbiome[94]. Here, we developed a broadly adaptable pipeline for modeling the metabolism of interacting organisms across physiologically distinct tissue (sub) sections. Using metabolic network reconstruction and constraint-based modeling, we studied the metabolism of a legume root nodule and SNF, a well-established model of inter-organismal metabolic exchange and cellular differentiation.

Our model (ViNE) accounts for plant shoot, root, and nodule tissues, with the nodule encompassing the metabolism of both the plant and bacterial partners and subdivided into five developmental zones. This is an advance over previous attempts at modeling SNF[33,35–40], most of which focused solely on bacterial metabolism while treating the plant as a black box. The increased complexity of ViNE allows for: i) more accurate simulations of the nutrient exchange, ii) analysis of the metabolic differentiation associated with nodule development, iii) examination of unexpected emergent properties of the symbiosis resulting from inter-organism interactions, and iv) the possibility to perturb the

network at the single reaction level. Initial simulations with ViNE supported that this model does a good job at capturing the metabolism of a legume nodule. Nevertheless, as with all models, ViNE predictions were imperfect; of the 38 genes both present in ViNE and listed as being involved in symbiosis according to Additional File 5 of Galardini et al.[95] (excluding genes involved in early nodulation, a process absent in ViNE), deletion of 84% were correctly predicted to have a symbiotic phenotype in ViNE. However, as we often compared simulated phenotypes for *M. truncatula* with experimental data for *M. sativa*, and given that rhizobium mutant phenotypes are often plant specific (e.g.[96–98]), we cannot rule out that some of the inconsistencies are the result of plant-specific phenotypes. Going forward, we intend to continue to manually refine and update ViNE to maximize consistency with experimental observations.

FBA simulations with ViNE revealed a pattern of diminishing returns in terms of plant growth (as a proxy for fitness) as the N₂-fixation efficiency increased, assuming that the rate of nodulation could also vary (Fig. 5c). This observation has potential implications for engineering SNF for biotechnological applications. It suggests that when developing rhizobial inoculants, maximizing competition for nodule occupancy may have a greater impact than maximizing the rate of N₂-fixation. This result also supports efforts aimed at engineering N₂-fixing symbiosis with cereals[99] by highlighting how even a low-efficiency symbiosis has the potential to have a noticeable benefit on crop yield.

At the same time, the pattern of diminishing returns is interesting from an evolutionary perspective[100]. In particular, the evolution of N₂-fixation efficiency may be influenced by the rhizobium community diversity, assuming that nodule infection increases the fitness of rhizobia[101]. In an environment dominated by a single rhizobium, kin selection may favor the evolution of a poorly efficient symbiosis as it would increase nodule number and thus the size of the niche for colonization by the rhizobia. On the other hand, in a highly diverse environment, evolution of strains capable of entering into a highly efficient symbiosis may be favored, as this would lead to fewer nodules and thus less plant resources being allocated to competing rhizobium strains, thereby limiting the spread of less mutualist (viz. cheater) strains[20,102].

Of particular interest to us were the metabolic exchanges between the plant and rhizobia, both during N₂-fixation and during differentiation. The carbon source(s) of differentiating rhizobia remain poorly understood. Results with ViNE suggested that sucrose may be a major carbon source for the differentiating bacteroids. However, *S. meliloti* mutants unable to transport sucrose are not impaired in nodule formation[103], suggesting that differentiating bacteroids have access to at least one other carbon source. Interestingly, a *S. meliloti pyc* mutant unable to grow with glycolytic carbon sources was not impaired in differentiation[60]. Similarly, *S. meliloti pckA*[58] and *tpi*[104] mutants unable to grow with gluconeogenic carbon sources remained capable of differentiating. Thus, it seems likely that differentiating bacteroids have access to a variety of glycolytic and gluconeogenic carbon substrate, with sugars possibly serving as the main carbon source in wild type nodules. If so, the restriction of carbon flow to N₂-fixing bacteroids to just $C_4$-dicarboxylates may be the result of active remodeling of the peribacteroid membrane during differentiation.

In attempting to identify conditions favoring the use of $C_4$-dicarboxylates as a carbon source by N₂-fixing bacteroids, ViNE also provided insights into the metabolic exchange in the N₂-fixation zone. The peribacteroid space of N₂-fixing bacteroids is known to be acidic due to the activity of H⁺-ATPases on the peribacteroid membrane[81–83]. This acidification contributes to the import of $C_4$-dicarboxylates and the export of ammonium from/to the plant cytosol and the peribacteroid space[105], and it

may contribute to the lysis of non-functional symbiosomes[106]. Our FBA simulations suggest that the plant-derived protons of the peribacteroid space may also be actively used by the bacteroid to support its metabolism.

Although it is generally accepted that nodules are low oxygen environments[84], the site of $O_2$-limitation has been debated. Based on the average concentration of free oxygen in the nodule, enzyme kinetics data are consistent with the mitochondria being $O_2$-limited and the bacteroids being $O_2$-sufficient[84–87]. Measurements of the adenylate pools of the plant and bacterial nodule fractions support this conclusion[90]. However, others have argued that nodule adenylate measurements suggest that bacteroids, not the plant, are $O_2$-limited[107]. Similarly, it was suggested that mitochondria cluster near the periphery of the cell near air pockets, resulting in elevated local $O_2$ concentrations[108,109]. The FBA results presented here predicted that $C_4$-dicarboxylates are the optimal carbon source for $N_2$-fixing bacteroids only when the plant mitochondria are $O_2$-limited while the bacteroids are $O_2$-sufficient (Fig. 6). This result supports the hypothesis that mitochondria, and not bacteroids, are $O_2$-limited in wild type nodules.

Although potentially powerful, the use of metabolic modeling to study SNF is not without limitations. In particular, the accuracy of predictions is restricted by the quality of the imposed flux constraints, and unfortunately, experimental kinetic data for key enzymes and nutrient exchange reactions in the nodule is generally lacking. FBA also fails to actively incorporate regulatory feedback control during simulations that could influence the metabolic properties of the nodule. Furthermore, the lack of finished legume genomes, an incomplete ability to ensure correct subcellular compartmentalization in eukaryotic cells, and difficulty in experimentally validating the functions of plant genes can limit the quality of models of SNF.

In sum, this work presented a complex metabolic model representing the full metabolism of a rhizobium-nodulated legume, as well as a series of simulations demonstrating the potential for this model to help address genetic, evolutionary, metabolic, and sociobiological questions. Future work will be aimed at continuing to refine and improve the quality of the model, and to using it to generate hypotheses to guide experimental studies and to assist in the interpretation of experimental datasets.

## Methods

**Construction of a virtual nodule environment.** Before starting the reconstruction of the integrated metabolic model, we refined/updated the existing reconstructions for *S. meliloti* and *M. truncatula*. Concerning *S. meliloti*, we started from the existing core metabolic reconstruction iGD726[34], which is a highly curated model with 726 genes and 681 reactions covering the pathways required to produce all biomass components when provided glucose, succinate, or sucrose as a carbon source, and made the following changes: (i) updated the biomass composition (Supplementary Data 7), (ii) manually added and curated accessory metabolic pathways, (iii) performed an automated expansion based on a slightly modified version of the iGD1575 model reported in[34], (iv) mass and charge balanced all reactions, and (v) added an ATP hydrolysis reaction to account for non-growth associated maintenance (NGAM) costs. The final *S. meliloti* reconstruction, termed iGD1348, is available in Supplementary Data 1 in SBML, XLS, and MATLAB COBRA format. In terms of the *M. truncatula* reconstruction, a recently published reconstruction[40] was used in this work. Since that version was based on the *M. truncatula* genome annotation version Mt3.5v5[42], the gene names in the reconstruction were updated to match the most recent annotation (version 5.0;[44] see Supplementary Note 2 for the details of this procedure, and Supplementary Data 8 for the model in SBML format). In addition, we included an energetic cost (0.25 mol of ATP per mol of transported compound) for all single-metabolite diffusion reactions to limit inappropriate transport between compartments. Next, a tissue-specific *M. truncatula* model containing shoot and root tissues was generated using the 'BuildTissueModel' function of Pfau et al.[40], all genes were relabelled according to the different model compartments (e.g., 'Root_'), and a few ion transport reactions were added. See Supplementary Notes 1 and 2 for additional details on preparation of the *S. meliloti* and *M. truncatula* GENREs, respectively.

Afterward, the *S. meliloti* and *M. truncatula* reconstructions were integrated following the procedure described in detail in Supplementary Notes 4–6, and all reactions producing dead-end metabolites were removed (Supplementary Data 9). This pipeline generated an integrated, multi-compartment metabolic reconstruction embedding seven overall compartments: shoot tissue, root tissue, the nodule apical meristem (nodule zone I) and four nodule developmental zones representing the combined metabolism of the plant and the bacterium (zone II distal, zone II proximal, interzone II-III, and the nitrogen-fixing zone III). Additionally: (i) the nodule compartments were metabolically connected to the root compartment using appropriate nutrient exchange reactions, (ii) the reaction space of each nodule zone was constrained based on the *M. truncatula – S. meliloti* zone-specific RNA-seq data of Roux and coworkers[50], which was reanalyzed as described in Supplementary Note 7, and (iii) biomass reactions were prepared for each compartment and the biomass of each compartment was combined using appropriate ratios estimated from the literature. Unless stated otherwise, the shoot and root tissue accounted for 98% of biomass (at a 2:1 ratio) and the nodule tissue collectively accounted for 2% of biomass. The final version of the integrated model was named ViNE; it is schematically represented in Fig. 2 and it is provided in Supplementary Data 10 as MATLAB COBRA and SBML formatted files.

**Metabolic modeling procedures.** Model integration, model manipulations, and FBA simulations were performed in MATLAB R2016b (mathworks.com) using the SBMLToolbox version 4.1.0[110], libSBML version 5.13.0[111], and scripts from the COBRA Toolbox commit 9b10fa1[112], the TIGER Toolbox version 1.2.0-beta[113], FASTCORE version 1.0[114], and the Tn-Core Toolbox version 2.2[48]. The iLOG CPLEX Studio 12.7.1 solver (ibm.com) was used for nearly all FBA simulations; the exception was for the preparation of iGD1348, during which the Gurobi version 7.0.1 solver (gurobi.com) was used. The switch to CPLEX was prompted by numerical issues that were solved by switching the solver. All custom scripts used in this study are available through a GitHub repository (github.com/diCenzo-GC/ViNE_Reconstruction).

Each gene found in multiple tissues or nodule zones was distinguished by a unique gene name to facilitate tissue-specific gene deletion analyses. When performing global single or double gene deletion analyses, all versions of the gene were simultaneously deleted followed by the removal of all constrained reactions. In contrast, zone- or tissue-specific gene deletion analyses involved deleting just the gene version specific to the zone or tissue of interest. Flux variability analyses were performed with the requirement that flux through the objective function was at least 99% the optimal flux. The robustness analyses involved first identifying the approximate flux range for each reaction in which the plant growth rate was non-zero. Then, for each reaction, the flux rate of the reaction was set to various values within the previously identified flux range, and the objective value was maximized. For simulations in which the rate of nodulation could vary, nodule biomass was removed from the objective reaction and instead forced through a nodule biomass sink reaction at the appropriate rate; maintenance costs and oxygen availability were modified accordingly (see Supplementary Note 8, for details). For simulations comparing the effect of providing zone III bacteroids sucrose versus $C_4$-dicarboxylates as the carbon source, a modified version of ViNE was prepared as described in Supplementary Note 3.

Except when stated otherwise, the objective function for all simulations was:

0.65333 Shoot biomass + 0.32666 Root biomass + 0.00100 Nodule Zone I biomass + 0.00675 Nodule Zone IId biomass + 0.00225 Bacteroid Zone IId biomass + 0. 00675 Nodule Zone IIp biomass + 0. 00225 Bacteroid Zone IIp biomass + 0.00075 Nodule Zone IZ biomass + 0.00025 Bacteroid Zone IZ biomass → 1.00 Biomass

Except when stated otherwise, the following constraints were placed on ViNE during the simulations: (i) when $N_2$-fixation was the sole source of nitrogen, the uptake of nitrate and ammonia by the roots was set to 0; (ii) the rate of oxygen uptake by zone III tissue was 8.9847 µmol h$^{-1}$ (g plant dry weight)$^{-1}$; (iii) the rate of light uptake by the leaves was 1000 µmol h$^{-1}$ (g plant dry weight)$^{-1}$; (iv) the rate of $CO_2$ uptake by the leaves was 1000 µmol h$^{-1}$ (g plant dry weight)$^{-1}$; (v) maintenance costs (ATP hydrolysis) for shoot and root tissue were ~74 and 37 µmol h$^{-1}$ (g plant dry weight)$^{-1}$, respectively; (vi) maintenance costs (ATP hydrolysis) for zone I, IId, IIp, IZ, and III plant nodule tissue were ~111, 750, 750, 83, and 833 nmol h$^{-1}$ (g plant dry weight)$^{-1}$, respectively; (vii) maintenance costs (ATP hydrolysis) for zone IId, IIp, IZ, and III bacteroid tissue were ~5.7, 5.7, 0.6, and 6.3 µmol h$^{-1}$ (g plant dry weight)$^{-1}$, respectively; and (vii) the only carbon sources available to bacteroids in the zone III nodule tissue was succinate, malate, and fumarate at a maximal rate of 1000 µmol h$^{-1}$ (g plant dry weight)$^{-1}$ each. No limit on the carbon sources available to the bacteroids of other nodule zones was set.

**Reporting summary.** Further information on research design is available in the Nature Research Reporting Summary linked to this article.

## Data availability

The authors declare that the data supporting the findings of this study are available within the paper (in Supplementary Data 1–10). No restrictions apply to data availability.

## Code availability

Code and scripts used to generate the integrated reconstruction are available at https://github.com/diCenzo-GC/ViNE_Reconstruction.

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

## Acknowledgements

GCD was supported by a postdoctoral fellowship from the Natural Science and Engineering Research Council of Canada, and funding from Queen's University. AM was supported by a grant from Fondazione Cassa di Risparmio di Firenze (project name: Metatrack). This work was partially supported by a grant from the Italian Ministry of Agriculture (MICRO4Legumes, act n. n.89267, 19-12-2019).

## Author contributions

G.D., A.M., and M.F. conceived the study. G.D., T.P., M.F. prepared the metabolic models. G.D., M.T., M.F. ran simulations with the model. G.D., A.M., and M.F. interpreted the results. G.D., A.M., and M.F. wrote the manuscript. All authors critically revised the manuscript and approved the final version.

## Competing interests

The authors declare no competing interests.
