## [Peer Review File · Nature Communications]

Reviewers' comments:

Reviewer #1 (Remarks to the Author):

Authors implement metabolic modeling to study symbiotic nitrogen fixation (SNF) that takes place between *Medicago truncatula* and *Sinorhizobium meliloti*. Overall, the manuscript is clearly written with a comprehensive list of relevant citations. Approaches taken are technically sound.

A downside is the little absence of technical innovation. Genome-scale metabolic models were well reconstructed, but were largely based on the previous studies. Considering the biological complexity associated with the SNF, it would have been nice to include additional information/approaches, for example relevant regulatory and signaling networks.

Following are some technical comments:

Introduction: Please briefly mention when genomes of *M. truncatula* and *S. meliloti* were sequenced for the first time.

Materials and methods: Please run Memote for the ViNE, and briefly state the results.

Line 128: Please provide more information on 'core' metabolic model. What pathways does it cover? What is its model size?

Figure 1: ViNE has many transport reactions between compartments, e.g., nodule and root compartments. Are the transport reactions relatively well known? What would be the confidence level for these transport reactions? Addition/absence of such transport reactions has profound effects on the simulated metabolic flux distribution.

Lines 198-207: Please explain why the final version iGD1348 has fewer genes than iGD1575. What is their main difference? This would help understand the descriptions regarding Figure 2.

Line 222: Please precisely define "Context-specific core metabolic models".

Lines 278 and 292: Please provide overall statistics (i.e., TP, TN, FP and FN) in comparison with published literature.

Discussion: Please discuss limitations of metabolic modeling when studying the SNF.

Minor comments

Line 86: "...require an understanding of what we know and what we don't know, ..." may be simplified as "...require our relevant knowledge, ..."?

Line 130: What is "iGD1575b"? Perhaps, "iGD1575"?

Line 131: "...charged balanced..." -> "...charge balanced..."

Reviewer #2 (Remarks to the Author):

Summary

This study is a computational exploration of symbiotic nitrogen fixation which is realized in the metabolic interactions between the legume plant *Medicago truncatula* and its rhizobial symbiotic bacterial partner *Sinorhizobium meliloti*. The bacteria invade and colonize *Medicago* roots and differentiate into forms that are specialized to fix atmospheric nitrogen. Fixed nitrogen is traded with the host plant against reduced carbon, which establishes a metabolic interaction of mutual benefit. Both *M. truncatula* and *S. meliloti* have fully sequenced genomes. Three interacting plant tissues and four different stages of internalized bacteria residing within nodules are represented as cellular metabolic models. The integrated plant model allows to study aspects of resource allocation between the plant and the rhizobia / symbionts.

First, the authors improved genome-scale metabolic reconstructions for *S. meliloti* and *M. truncatula*, which were then used to make leaf, root, and nodule sub-models into an integrated model called Virtual Nodule Environment (ViNE). Simulated variations in N₂ fixation efficiency and/or rate of nodulation showed there were diminishing returns in plant growth rate increases due to improvements in fixation efficiency, and that the plant benefitted from lower nodulation rates (lower biomass proportion) because they did not need to invest in nodule maintenance costs. Differences in energy production and other metabolic activities associated with nodule development were identified by comparing reaction essentialities in nodule zones specific for uninfected cells, differentiation, and nitrogen fixation. Further examination of the nitrogen fixation zone led to the finding that limitations in mitochondrial terminal oxidase activity could be responsible for C₄-dicarboxylates being traded to the bacterial partner from the plant instead of sucrose.

1. Altogether the integrated model is a very complex multicellular set-up. The authors put much emphasis on detailing the nodule into 5 different cell types / zones. However, the motivation to do this could be clearer. Which literature background could be cited to justify differentiating these zones?

2. The simulations shown in Figure 5 are explained in the supplement (Text S5), but the very complex iterative optimization procedures are hard to conceptualize. Could there be a supplemental figure to help understanding this?

In the main text, some details on description could be made clearer:

There seems to be some confusion with regards to how biomass ratios are referred to. There is "plant to nodule ratio" and "rate of nodulation". These definitions are the same? If yes, they could be clearer standardized throughout the manuscript. (In line 383: "We next evaluated the relationship between the rate of N₂-fixation (without modifying the plant to nodule ratio) ...". Supplemental Text S5 then talks about "rate of nodulation", which is "gram of nodule per gram of plant".)

Figure 5A: Here biomass formation (growth rate) is shown in response to an increasing rate of N₂ fixation forced onto the model. Is N₂ the only possible nitrogen source? After the point where the biomass production (growth rate) doesn't increase further, where does the reduced nitrogen end up? Are there effluxes allowed, like NH₄ or organic forms of nitrogen? If yes, does this make physiological sense that N₂ fixation products can leak out? Also, the text says that in figure 5A the ratio of nodule biomass to plant biomass is kept constant. The value used could be shortly mentioned in the text. Is it 98:2 as mentioned as "default" in the supplement? 98:2 would mean 2% nodule biomass. Is 2 % where the growth rate reaches its maximum in figure 5B?

Figure 5C: N₂ fixation efficiency. Figure 5C defines N₂ fixation efficiency as "rate of N₂-fixation per gram nodule". In Figure 5B this ratio is held constant? What's the value? The legend in Figure 5C spell out N₂ fixation efficiency differently: "... with a constant rate of N₂-fixation per gram of nodule)." For better understanding, again, one definition could be used throughout.

3. The objective function(s) and model constraints (e.g., light uptake, carbon metabolite able to be traded) should be declared in the main text
4. Lines 261-263 – What fixed rates of nodulation and nitrogen fixation were used here? How do they compare to literature findings?
5. Lines 261-263- The nodulation rate should be defined here
6. Table 2, third row: if the nitrogen source is "N₂-fixation" (i.e. N₂, not ammonium) and the nodulation State is "Non-nodulated" (i.e. no symbiotic bacteria present), how can the plant grow?
7. Lines 450 – 460 discuss metabolic consequences of limitations of free oxygen in the N₂ fixation zone. Helpful context for the reader might be here to mention the oxygen sensitivity of nitrogenase, which dictates the physiological low oxygen conditions in the N₂ fixation zone.

FURTHER COMMENTS

7. Line 30 – "protons transfer protons"
8. Line 121 – "will to"
9. Table 1 – why are the model properties of Zone III not listed?
10. Table S1 – What is number/units of "trace" metabolites/elements for the model?
11. Figure S3 – The legend refers to Figure 5D, but there is no Figure 5D; Was Figure 5C to be referenced? Actually, Figures 5C and S3 look identical. Is this the case?

Reviewer #1 (Remarks to the Author):

Authors implement metabolic modeling to study symbiotic nitrogen fixation (SNF) that takes place between *Medicago truncatula* and *Sinorhizobium meliloti*. Overall, the manuscript is clearly written with a comprehensive list of relevant citations. Approaches taken are technically sound.

A downside is the little absence of technical innovation. Genome-scale metabolic models were well reconstructed, but were largely based on the previous studies. Considering the biological complexity associated with the SNF, it would have been nice to include additional information/approaches, for example relevant regulatory and signaling networks.

Response: We thank the Reviewer for commenting that the metabolic models were well reconstructed. As noted by the Reviewer, these models were developed from existing *S. meliloti* and *M. truncatula* metabolic models previously constructed by authors on the current manuscript. Nevertheless, the process of the multicomponent integration of the models of these two species goes well-beyond the existing models for these organisms, which we feel is an important technical innovation in its own right. In addition, the process of constraining the reaction space of ViNE was guided by RNA-seq data, meaning that ViNE inherently contains some expression data. That said, we agree that integration of additional data levels on the model, such as regulatory or signaling networks, would be an interesting direction for future research but we feel that it goes beyond the scope of the current manuscript (being the generation of a FBA model of a nodulated legume). Well-detailed data on signaling in nodules are present in the literature (see for instance recent papers as Mergaert *et al.* Plant Cell 2020, 32: 42-68 and Carrère *et al.* 2020 Plant Cell Physiol 61:203–211 with the LeGOO database). However, these studies mostly describe the initiation and control of differentiation, nodule organogenesis, and infection, with little evaluation of the control of metabolic shifts. As such, there is currently little data on the signalling or regulatory networks that would be useful for integration with a FBA model.

Following are some technical comments:

Introduction: Please briefly mention when genomes of *M. truncatula* and *S. meliloti* were sequenced for the first time.

Response: Done. The years that the genomes were published have been added to the introduction at lines 98-99.

Materials and methods: Please run Memote for the ViNE, and briefly state the results.

Response: Done. Memote has been run on ViNE, as well as the stand-alone plant and bacterial reconstructions. Memote reports have been included as Supplementary Materials S5 in the new version of the manuscript. Overall, when Memote was run on ViNE, a total confidence score of 72% was obtained. To put this value in perspective, we also ran Memote on the two models that were used to build ViNE. The *S. meliloti* and *M. truncatula* models scored 78% and 65%, respectively, thus revealing that the integrated reconstruction score is comparable (or even better) than the original ones. These results are briefly stated at lines 253-258 of the new version of the manuscript and details on how Memote was launched are provided as Supplementary Material S1,

Text S5.

Line 128: Please provide more information on ‘core’ metabolic model. What pathways does it cover? What is its model size?

Response: Done. As suggested, we have added the following information at lines 117-120: “... we started from the existing core metabolic reconstruction iGD726³⁴, which is highly curated model with 726 genes and 681 reactions covering the pathways required to produce all biomass components when provided glucose, succinate, or sucrose as a carbon source, ...”

Figure 1: ViNE has many transport reactions between compartments, e.g., nodule and root compartments. Are the transport reactions relatively well known? What would be the confidence level for these transport reactions? Addition/absence of such transport reactions has profound effects on the simulated metabolic flux distribution.

Response: As the Reviewer notes, there are many reactions moving compounds between compartments, and the transporters that are included are likely to have an impact on the downstream simulations. Unfortunately, there is not much information on the transfer of metabolites between these tissues in the literature, and thus it is likely that the model will have to be updated as new information is obtained (as is true for all existing metabolic models for all organisms). Thus, we chose to take a relatively unrestrictive approach at first, and then allow the integration of the RNA-seq expression data to guide the selection of which transfer reactions would be active by turning on/off metabolic pathways that would act on the transferred metabolites.

The metabolites allowed to move between the root and shoot tissues were taken from the previously published *M. truncatula* model (ref 40 in the updated manuscript), and so we assume them to be reasonable. The metabolites allowed to move between the root and nodule include the same metabolites that can transfer between the root and shoot (if these metabolites are moving between the root and shoot, they must be in the phloem and thus also able to move into the nodule tissue) and anything that can be imported by the root from the soil (as we assume the nodule tissue can transport the same compounds). As little is known about the transfer of metabolites between the plant cytosol and the bacteroids in the nodule, especially outside of zone III, we were unrestrictive; any compound in the cytosol for which *S. meliloti* has a transporter was allowed to be transported (at least prior to the integration of the RNA-seq expression data).

To test the impact of the inclusion or absence of individual reactions for the between-tissue transfer of metabolites, we tested the effect of individually deleting each of these reactions on the overall rate of plant growth in the unconstrained model prior to integration of RNA-seq data. Of the 110 reactions to move metabolites between the root, shoot, or nodule, the deletion of nearly half (53 reaction) had less than a 5% impact on overall plant growth. Of the 57 with an impact, all of them represent either key metabolites expected to be transferred (e.g., sucrose) or essential nutrients or micronutrients (e.g., sulfate, metal ions) that could only be imported from the external environment by only one tissue and are thus expected to be transferred to the other tissues. Of the 768 initial reactions for the transfer of metabolites between the plant and bacteria in the nodule, the deletion of only 53 (7%) had more than a 5% impact on overall plant growth. These 53 reactions transport essential micronutrients and minerals and gases like oxygen and nitrogen, and thus are expected to be required. Overall, these simulations support that the specific choice of between-tissue transfer reactions has little impact on the overall plant growth rate (the main output

considered in this work), although we cannot rule out an effect on flux distribution.

We have now included this information as Supplementary Materials S1, Text S4.

Lines 198-207: Please explain why the final version iGD1348 has fewer genes than iGD1575. What is their main difference? This would help understand the descriptions regarding Figure 2.

Response: Done. As suggested by the Reviewer, the following information has been added at lines 214-216: “The reduced size of iGD1348 relative to the older iGD1575 model is a consequence of deleting poorly characterized genes that appeared to be incorrectly associated with reactions, and the removal of pathways that produced dead-end metabolites.”

Line 222: Please precisely define “Context-specific core metabolic models”.

Response: Done. As suggested by the Reviewer, we have modified this sentence (now lines 231-233) to read as follows: “Context-specific core metabolic models, containing a minimal set of reactions for biomass production in a given environment, were extracted from the iGD1348 and iGD1575 metabolic models through the integration of Tn-seq data³⁴ using Tn-Core⁵²”.

Lines 278 and 292: Please provide overall statistics (i.e., TP, TN, FP and FN) in comparison with published literature.

Response: Partially done. We feel that the appropriate experimental data to provide proper TP, TN, FP, and FN values are not available in the literature. There is only experimental symbiotic data for a minority of *S. meliloti* and *M. truncatula* genes, limiting the size of the comparison that can be done. In addition, the main plant model used when testing *S. meliloti* mutants for symbiotic phenotypes is alfalfa, not *M. truncatula* (but *M. truncatula* is the main plant genetic model and has a sequenced genome, unlike alfalfa, which is why alfalfa was not used to make ViNE). It is common for mutations to have a plant-specific phenotype, with a mutation impairing symbiosis with one plant but not with a second plant (e.g., ref 100 in the updated manuscript); thus, most of the available data is for *S. meliloti* in symbiosis with alfalfa not with *M. truncatula*. Thus, even for genes with data, it is not possible to state whether discrepancies are due to incorrect model predictions or due to differences in plant species. Therefore, we have not provided overall statistics in the results section. However, we instead added some of this information to the discussion within the paragraph where we briefly touched upon these issues. Lines 472-480 now read as: “Nevertheless, as with all models, ViNE predictions were imperfect; of the 38 genes listed as being involved in symbiosis according to Additional File 5 of⁹⁹ (excluding genes involved in early nodulation, a process absent in ViNE) that are also present in ViNE, deletion of 84% were predicted to have a symbiotic phenotype in ViNE. However, as we often compared simulated phenotypes for *M. truncatula* with experimental data for *M. sativa*, and given that rhizobium mutant phenotypes are often plant specific [e.g.,¹⁰⁰⁻¹⁰²], we cannot rule out that some of the inconsistencies are the result of plant-specific phenotypes. Going forward, we intend to continue to manually refine and update ViNE to maximize consistency with experimental observations.”

Discussion: Please discuss limitations of metabolic modeling when studying the SNF.

Response: Done. We agree that a discussion of the limitations is important. We have now added

this information at lines 532-539 of the discussion.

Minor comments

Line 86: "...require an understanding of what we know and what we don't know, ..." may be simplified as "...require our relevant knowledge, ..."?

Response: Done. We modified this statement to read: "... require an in-depth understanding of the symbiotic interaction ..." (now lines 73-74)

Line 130: What is "iGD1575b"? Perhaps, "iGD1575"?

Response: Done. iGD1575b was reported in our previous publications (diCenzo et al. 2018. PLOS Genetics. 14:e1007357), and represents a slightly modified version of our iGD1575 model. To clarify this, we have reworded this statement (now at lines 122-123) as follows: "performed an automated expansion based on a slightly modified version of the iGD1575 model reported in ³⁴"

Line 131: "...charged balanced..." -> "...charge balanced..."

Response Done. We thank the Reviewer for catching this error. It has now been corrected. (now line 123)

Reviewer #2 (Remarks to the Author):

Summary

This study is a computational exploration of symbiotic nitrogen fixation which is realized in the metabolic interactions between the legume plant *Medicago truncatula* and its rhizobial symbiotic bacterial partner *Sinorhizobium meliloti*. The bacteria invade and colonize *Medicago* roots and differentiate into forms that are specialized to fix atmospheric nitrogen. Fixed nitrogen is traded with the host plant against reduced carbon, which establishes a metabolic interaction of mutual benefit. Both *M. truncatula* and *S. meliloti* have fully sequenced genomes. Three interacting plant tissues and four different stages of internalized bacteria residing within nodules are represented as cellular metabolic models. The integrated plant model allows to study aspects of resource allocation between the plant and the rhizobia / symbionts.

First, the authors improved genome-scale metabolic reconstructions for *S. meliloti* and *M. truncatula*, which were then used to make leaf, root, and nodule sub-models into an integrated model called Virtual Nodule Environment (ViNE). Simulated variations in N₂ fixation efficiency and/or rate of nodulation showed there were diminishing returns in plant growth rate increases due to improvements in fixation efficiency, and that the plant benefitted from lower nodulation rates (lower biomass proportion) because they did not need to invest in nodule maintenance costs. Differences in energy production and other metabolic activities associated with nodule development were identified by comparing reaction essentialities in nodule zones specific for uninfected cells, differentiation, and nitrogen fixation. Further examination of the nitrogen fixation zone led to the finding that limitations in mitochondrial terminal oxidase activity could be responsible for C₄-dicarboxylates being traded to the bacterial partner from the plant instead of

sucrose.

1. Altogether the integrated model is a very complex multicellular set-up. The authors put much emphasis on detailing the nodule into 5 different cell types / zones. However, the motivation to do this could be clearer. Which literature background could be cited to justify differentiating these zones?

Response: Done. Pioneering studies, such as those by Vasse *et al.* 1990. J Bacteriol 172:4295, demonstrated that legume nodules with an indeterminate structure, such as those formed by *M. truncatula*, contain five spatially-distinct zones: zone I, zone II, interzone II-III, zone III, and zone IV. We did not include zone IV in our model as this is the zone of senescence. Instead, we split zone II into zone II distal and zone II proximal. The decision to do so was to match the zones chosen by Roux *et al.* 2014. Plant J 77:817, who performed laser capture microdissection coupled to RNA-seq, as we wished to use their RNA-seq data in the construction of ViNE. However, as note by the Reviewer, this was poorly explained in the manuscript. We have now added the following statement at lines 260-262 of the Results: “The nodule was subdivided into five zones to match the spatially, and transcriptionally, distinct developmental zones that are simultaneously present in indeterminate legume nodules such as those formed by *M. truncatula*^{21,46}.”

2. The simulations shown in Figure 5 are explained in the supplement (Text S5), but the very complex iterative optimization procedures are hard to conceptualize. Could there be a supplemental figure to help understanding this?

Response: Done. We agree with the Reviewer that it is difficult to conceptualize the process used to generate the data for Figure 5. In the hopes of improving this, we have now added a flow chart illustrating this optimization procedure, and one other, as new Supplementary Figures (Figures S4 and S5 in Supplementary Materials S1). Additionally, an error in the description of this procedure in Text S5 (now Text S7) was fixed.

In the main text, some details on description could be made clearer:

There seems to be some confusion with regards to how biomass ratios are referred to. There is “plant to nodule ratio“ and “rate of nodulation”. These definitions are the same? If yes, they could be clearer standardized throughout the manuscript. (In line 383: “We next evaluated the relationship between the rate of N₂-fixation (without modifying the plant to nodule ratio) ...”. Supplemental Text S5 then talks about “rate of nodulation”, which is “gram of nodule per gram of plant”.)

Response: Done. Indeed, these two terms were referring to the same thing. We have therefore removed the phrase “plant to nodule ratio” and used “rate of nodulation” throughout.

Figure 5A: Here biomass formation (growth rate) is shown in response to an increasing rate of N₂ fixation forced onto the model. Is N₂ the only possible nitrogen source?

Response: Done. Yes, N₂ is the sole source of nitrogen in all panels of Figure 5; we have now stated this in the Figure 5 legend. In addition, the units on the X-axis were updated to maintain

consistency throughout the manuscript.

After the point where the biomass production (growth rate) doesn't increase further, where does the reduced nitrogen end up? Are there effluxes allowed, like NH₄ or organic forms of nitrogen? If yes, does this make physiological sense that N₂ fixation products can leak out?

Response: Done. As the Reviewer suggests, we allowed the excess fixed nitrogen to be exported as ammonia to be able to run these analyses. We chose to allow the export of ammonia for simplicity, but we expect the same results if we instead allowed the export of amino acids. The ability of nitrogen to be exported from plant root tissue makes sense physiological, as efflux of nitrogenous compounds including ammonia and nitrate from plant roots can occur (Glass *et al.* 2001. *J Plant Nutr Soil Sci* 164:199). Biologically, however, we agree (and the simulations support) that it would make little sense to waste energy on N₂-fixation if the product is exported and thus it is unlikely to occur much in nature. However, there are some data that suggest at least a small amount of fixed nitrogen may be lost from the nodule/root systems through N₂O emission and rhizodeposition (Ofosu-Budu *et al.* 1992. *Soil Sci Plant Nutr* 38:717; Rochette & Janzen. 2005. *Nutr Cycling Agroecosyst* 73:1711 Fustec *et al.* 2010. *Agron Sustain Dev* 30:57; Paynel *et al.* 2001. *Plant Soil* 229:235), while experiments with N¹⁵ demonstrated and measured the amount of fixed nitrogen that is excreted by the roots (Ta *et al.* 1986. *Can J Bot* 64:2063).

Also, the text says that in figure 5A the ratio of nodule biomass to plant biomass is kept constant. The value used could be shortly mentioned in the text. Is it 98:2 as mentioned as “default” in the supplement? 98:2 would mean 2% nodule biomass. Is 2 % where the growth rate reaches its maximum in figure 5B?

Response: Done. As suggested by the Reviewer, we have now noted that the ratio is 98:2 (now phrased as a 2% rate of nodulation) in the Figure 5 legend and in the Materials and Methods (lines 152-153). That is the correct interpretation, at a 98:2 ratio, 2% of the total weight of the plant would be nodule tissue – we no longer use the phrase “plant to nodule ratio” and instead use “rate of nodulation”; as a result, we no longer say a 98:2 ratio and instead say 2%. We have clarified this in the Materials and Methods (lines 152-153). In Figure 5B, the growth rate reaches its maximum at 2.05%.

Figure 5C: N₂ fixation efficiency. Figure 5C defines N₂ fixation efficiency as “rate of N₂-fixation per gram nodule”. In Figure 5B this ratio is held constant? What's the value? The legend in Figure 5C spell out N₂ fixation efficiency differently: “... with a constant rate of N₂-fixation per gram of nodule.” For better understanding, again, one definition could be used throughout.

Response: Done. Yes, the N₂-fixation efficiency was kept constant during the simulations shown in Figure 5B, and this (and its value) is now stated in the Figure 5 legend. As suggested by the Reviewer, we have standardized the terms N₂-fixation efficiency and rate of nodulation throughout the manuscript and defined them at first use.

MINOR COMMENTS

3. The objective function(s) and model constraints (e.g., light uptake, carbon metabolite able to be

traded) should be declared in the main text

Response: Done. We agree with the Reviewer that this information is important and should be included in the main text. We have now added this information at the end of the Materials and Methods section (lines 184-201). We also clarified the bounds on the bacteroid maintenance costs in Text S3.

4. Lines 261-263 – What fixed rates of nodulation and nitrogen fixation were used here? How do they compare to literature findings?

Response: Done. We agree with the reviewer, and we have added the information on rates of nodulation and nitrogen fixation. (now lines 271-276). We have also added information on how the predicted rates of N₂-fixation compare with experimental data; however, this was done several lines earlier at lines 265-268.

5. Lines 261-263- The nodulation rate should be defined here

Response: Done. We agree with the reviewer, and we have defined the “rate of nodulation” here. (now lines 273-274).

6. Table 2, third row: if the nitrogen source is “N₂-fixation” (i.e. N₂, not ammonium) and the nodulation State is “Non-nodulated” (i.e. no symbiotic bacteria present), how can the plant grow?

Response: Done. As noted by the Reviewer, we failed to properly explain this. Nitrogen fixation and nodulation in ViNE, at least under standard running parameters, are unlinked from the computational perspective. The reactions required to carry flux for nitrogen fixation do not require flux through reactions needed to produce nodule biomass. In other words, it is possible for ViNE to fix nitrogen without producing nodule biomass. Of course, biologically this does not make sense, but computationally it does. We wished to examine how different aspects of the symbiosis contribute to the observed costs (such as the contribution of forming and maintaining nodule versus nitrogen fixation). Therefore, in the third row, the data comes from a simulation where the sole source of nitrogen for the plant was through nitrogen fixation, but nodule biomass was excluded from the objective function and nodule maintenance costs were removed. We have now clarified this in Table 2 by stating: “Non-nodulated indicates that nodule biomass was excluded from the overall biomass reaction and nodule maintenance costs were removed. As N₂-fixation in ViNE is generally unlinked to the production or maintenance of nodule biomass, N₂-fixation can occur nodulation.”

7. Lines 450 – 460 discuss metabolic consequences of limitations of free oxygen in the N₂ fixation zone. Helpful context for the reader might be here to mention the oxygen sensitivity of nitrogenase, which dictates the physiological low oxygen conditions in the N₂ fixation zone.

Response: Done. We agree with the Reviewer that this would be helpful context for readers. We have therefore modified this section (lines 426-431) to contain the following: “ViNE contains a limit on the rate of oxygen uptake by zone III nodule tissue as the concentration of free oxygen in the N₂-fixation zone is known to be low⁸⁸ to protect the highly oxygen-sensitive nitrogenase

enzyme. The expected consequence of this is a constriction in the rate of flux through the electron transport chain, thus, restricting nodule and bacteroid metabolism by limiting ATP production and the removal of reductant.”

FURTHER COMMENTS

7. Line 30 – “protons transfer protons”

Response Done. We thank the Reviewer for catching this error. It has now been corrected to “proton transfer”. (still line 30)

8. Line 121 – “will to”

Response Done. We thank the Reviewer for catching this error. It has now been corrected to “will”. (now line 110)

9. Table 1 – why are the model properties of Zone III not listed?

Response Done. We thank the Reviewer for catching this error as the properties for Zone III should have been included. We have updated Table 1, and it now contains the model properties for Zone III.

10. Table S1 – What is number/units of "trace" metabolites/elements for the model?

Response Done. As suggested, we have now indicated the values for the trace metabolites/ions in Table S1.

11. Figure S3 – The legend refers to Figure 5D, but there is no Figure 5D; Was Figure 5C to be referenced? Actually, Figures 5C and S3 look identical. Is this the case?

Response Done. We thank the Reviewer for catching this error. As noted, this should refer to Figure 5C, and this has been changed in the text. Figure 5C and S3 are very similar figures, but identical. In Figure 5C, there was no limit on the oxygen uptake rate in zone III; in Figure S3, an oxygen uptake rate was included. In general, this had little effect on the results, except for the rightmost datapoint in the figures (in Figure S3 it drops a bit, whereas this does not occur in Figure 5C). The purpose of included Figure S3 is to confirm that the presence/absence of an oxygen usage limit did not impact the pattern of the results.

Additional changes

A comparison of the predicted carbon costs of supporting the symbiotic interaction with experimentally determined values was added at lines 268-270.

The scale on the X-axes of Figure 5C and S3 were updated to correct an error.

REVIEWERS' COMMENTS:

Reviewer #1 (Remarks to the Author):

The reviewer's comments were all successfully addressed. Following are minor comments.

Line 474: Authors might want to add first author's name before the reference number "99" in the "...Additional File 5 of 99...". Please do the same for similar expressions that appear throughout the manuscript.

Ref. 55: A work on Memote has just been published in Nature Biotechnology (PMID: 32123384). Please update the ref. 55 accordingly.

Reviewer #2 (Remarks to the Author):

I see all the comments being made from 1st review well addressed.

REVIEWERS' COMMENTS:

Reviewer #1 (Remarks to the Author):

The reviewer's comments were all successfully addressed. Following are minor comments.

Line 474: Authors might want to add first author's name before the reference number "99" in the "...Additional File 5 of 99...". Please do the same for similar expressions that appear throughout the manuscript.

Authors' reply: Correction performed.

Ref. 55: A work on Memote has just been published in Nature Biotechnology (PMID: 32123384). Please update the ref. 55 accordingly.

Authors' reply: we have added the relevant reference.

Reviewer #2 (Remarks to the Author):

I see all the comments being made from 1st review well addressed.